# Pumping Well Layout Scheme Design and Sensitivity Analysis of Total Critical Pumping Rates in Coral Island Based on Numerical Model

**Ran Wang [1,2], Longcang Shu [1,2,*], Yuxi Li [1,2] and Portia Annabelle Opoku [1,2]**

[1] College of Hydrology and Water Resources, Hohai University, Nanjing 210098, China; 180201010010@hhu.edu.cn (R.W.); liyuxi@hhu.edu.cn (Y.L.); portia_a_opoku@hhu.edu.cn (P.A.O.)

[2] State Key Laboratory of Hydrology-Water Resources and Hydraulic Engineering, Hohai University, Nanjing 210098, China

\* Correspondence: lcshu@hhu.edu.cn; Tel.: +86-138-5194-1641

**Abstract:** Groundwater on small coral islands exists in the form of freshwater lenses that serve as an important water resource for local inhabitants and ecosystems. These lenses are vulnerable to salinization due to groundwater abstraction and precipitation variation. Determination of the sustainable yield from freshwater lenses is challenging because the uncertainties of recharge and hydrogeological characteristics make it difficult to predict the lens response to long-term pumping. In this study, nine pumping well layout schemes along a line are designed using the orthogonal experimental design method, and an optimal well layout scheme is determined by multi-index range analysis and comprehensive balance analysis method. The total critical pumping rates of the freshwater lens corresponding to different schemes are calculated by numerical simulation, and the sensitivity of the total critical pumping rates to hydrogeological parameters is analyzed. The results show that the calculation of the total critical pumping rates needs to be combined with the specific well layout scheme with consideration to the length of well screens, the number of wells and the distance between wells. The difference in total critical pumping rates between different schemes can be up to three times. The uncertainty of hydrogeological parameters has a great impact on the total critical pumping rates. Within the range of a 30% reduction in parameters, $\alpha$ and $K$ are the key risk factors of pumping; within the range of a 30% increase in parameters, $\alpha$, $n_e$ and $K$ are the key risk factors; $\alpha$-$n_e$ combined changes had the greatest impact. The management of freshwater lenses and the assessment of sustainable yield will continue to be important tasks for coral islands in the future, and this study can help with the sustainable exploitation of island freshwater lenses.

**Keywords:** freshwater lens; coral island; sustainable exploitation; sensitivity analysis

## 1. Introduction

The freshwater lens also called the Ghyben–Herzberg lens, was discovered by Ghyben [1] and Herzberg [2] in a freshwater supply study of the coastal areas of Europe in 1889 and 1901, respectively. Beneath an island, fresh groundwater may appear in the form of a "freshwater lens", which covers the seawater in a high permeability aquifer. Freshwater lenses are limited renewable groundwater resource on islands. Coral islands and atolls represent a specific subgroup of the broader category of islands. Bryan [3] listed 408 atolls spread over the oceans of the world, 294 in the Pacific Ocean, and 73 in the Indian Ocean, with the remainder distributed across the globe.

Coral islands are usually characterized by their dual-aquifer configuration. Unconformably deposited above the Pleistocene limestone reef deposit are unconsolidated or poorly consolidated Holocene sediments, primarily consisting of coral sand [4,5]. Pauw, P.S., et al. [6] used analytical solutions as a computationally faster alternative to numerical simulations to quantify saltwater coning below horizontal wells in freshwater lenses. The

hydraulic conductivity of coral sand is usually between 0.023–110 m/d, and the porosity and specific yield are usually between 0.4–0.55 and 0.012–0.310, respectively, which show that coral sand has the characteristics of higher water capacity but lower specific yield compared with terrigenous sand with the same particle size range [7].

Research on freshwater lenses has been conducted using numerical simulation and physical experiments, and these results have enhanced the understanding of groundwater resources management in coral islands. Bailey et al. [8] implemented Ayers and Vachers' [4] inclusive conceptual model for atoll island aquifers in a comprehensive numerical modeling study, to evaluate the response of the freshwater lens to selected controlling climatic and geologic variables. Chui and Terry [9] simulated the damage and recovery of the freshwater lens of the atoll after storm wave scouring. Davood et al. [10] used 2D and 3D simulation methods to model the freshwater lens of Kish Island in Iran's Persian Gulf arid area and discussed the dynamics of freshwater lenses and the potential threat of climate-induced seawater intrusion. Stofberg et al. [11] employed numerical simulation to give an appraisal of the risk of freshwater lenses disappearing in the Dutch coastal regions. Vincent et al. [12] developed a script-based numerical groundwater model of a freshwater lens to evaluate the potable water supply management scenarios. Briggs et al. [13] used numerical simulation to research the driving factors of freshwater lens dynamics observed in Palmyra Atoll National Wildlife Refuge from 2008 to 2019. Numerical models have also been used to research the formation, evolution, and dynamic process of freshwater lenses in coral islands [7,14–16]. Physical experiments have also been carried out to simulate the formation and exploitation of freshwater lenses [17–19]. A study on optimization of pumping rate and recharge to a small coral island aquifer conducted by Pallavi Banerjee and V.S Singh [20] showed that a true reflection of lens response throughout the year could only be accomplished by accounting for seasonal variability of aquifer recharge. This study also demonstrated the feasibility of using numerical models to manage fresh groundwater on islands. There is also research using real experiments on coastal aquifers. Comte J C et al. [21] focused on the hydrogeological outcomes of the research, framed within the principal socio-environmental issues identified. Bourhane A et al. [22] revealed a strong potential for both further developing coastal aquifers and initiating the prospection of inland aquifers.

While the role of real experiments on coastal aquifers has existed, there is still a study deficit on the impact of pumping on freshwater lenses, especially withoptimal well layouts and the uncertainty of hydrogeological parameters. Therefore, research is still needed to guide the management of freshwater lenses to ensure the provision of appropriate quality water.

In this study, a numerical model coupled with water flow and solute transport is utilized to simulate the up-coning caused by pumping in a coral island. An optimal well layout scheme is determined by the orthogonal experimental design method. The total critical pumping rates of the freshwater lenses are calculated by numerical simulation, and the sensitivity of the total critical pumping rates to hydrogeological parameters is analyzed. The objective of this paper is to serve as a resource for the development and utilization of island freshwater lenses, which have important practical applications.

## 2. Study Area

The coral island considered in this study is located to the southeast of Hainan Island and is the largest in the Xisha Archipelago, South China Sea, with a total area of 2.1 km$^2$. The climate is tropical marine monsoon, hot and humid, with an annual average temperature of 26.5 °C. The annual average precipitation of 1505 mm is sufficient; however, the seasonal distribution is irregular. In a year, there are apparent dry and wet seasons. The rainy season lasts from June to November, accounting for 80% of the annual rainfall; whereas the dry season is usually from December to May.

The topography is low and flat, with elevations of less than 6 m. It is difficult to form surface water due to the gentle slope of the terrain and the high permeability of the

surface sediments. Since the middle Holocene, the reef flat, beach, sand dike, sand mat, and depression have been formed under different sedimentary dynamic conditions. The depressions in the middle of the island are 2–3 m lower than the surrounding sandbank, which are developed by lagoons. The sand mat is the main body of the island, slightly higher than the depression, with flat terrain. The sand dike is surrounded by the island and is formed by the debris sand of coral shells. The narrow beach slopes to the sea. The reef flat, which is much larger than the island, is hidden under the sea.

The depth of groundwater in the coral island is about 0.3–2.9 m. Geological profiles based on 10 boreholes showed that the subsurface can be divided into a Holocene and an underlying Pleistocene unit. The contact between the Holocene and underlying Pleistocene sediments, known as the "Thurber Discontinuity" or the "Holocene-Pleistocene Unconformity" (HPU) [23], occurs at a depth about 20 m below the ground surface. The Pleistocene coral reef limestone is characterized by well-developed pores and dissolution cavities, high permeability, and easy seawater circulation. The Holocene granular sediments are mainly composed of coral clastic medium sand and coral clastic gravel sand. The hydraulic properties of the Holocene and Pleistocene units of the coral island are obviously different. The hydraulic conductivity ($K$) of the Pleistocene limestone is significantly higher than that of the Holocene sediments. Lens thickness for atoll islands in the Pacific and the Indian Ocean generally ranges from a few meters to 20 m, limited by the Thurber Discontinuity, as any freshwater below the discontinuity becomes salinized [24]. The maximum thickness of the freshwater lens measured on the study island is about 8–13.5 m.

A conceptual sketch of the coral island is shown as a diagram in Figure 1. The diagram includes a simplified geological setting and shows the dual nature of the aquifer system that results from the Holocene sediments overlying the Pleistocene karstified limestone. The recharge source of the groundwater is limited, and precipitation is the only natural source.

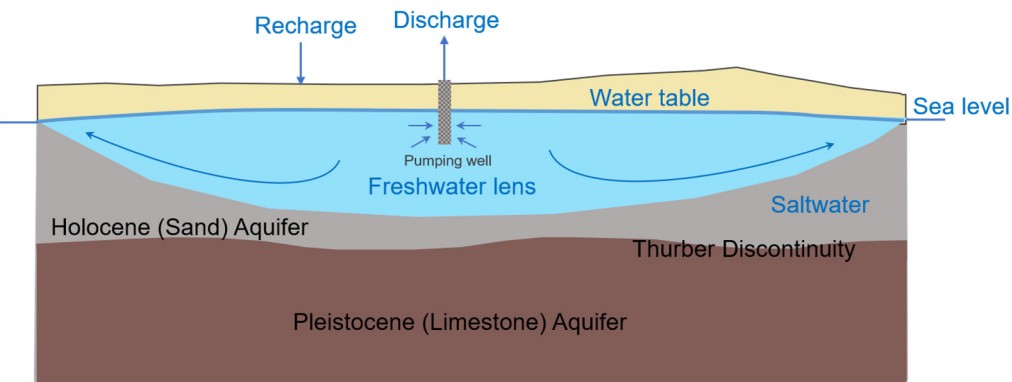

**Figure 1.** Conceptual sketch of a freshwater lens in a small coral island.

## 3. Materials and Methods

### 3.1. Numerical Model

A two-dimensional numerical density-dependent groundwater flow and solute transport model is developed to represent the profile of a coral island using SEAWAT [25], which can effectively reduce the computational workload and improve the efficiency of calculation. The source code for SEAWAT was developed by combining MODFLOW and MT3DMS into a single program that solves the coupled flow and solute-transport equations. MODFLOW was modified to solve the variable-density flow equation by reformulating the matrix equations in terms of fluid mass rather than fluid volume, and by including the appropriate density terms. This interpretative model is used to simulate the dynamics of freshwater lenses of coral islands with similar hydrogeological conditions. The model is run for 100 years, during which time the freshwater lens develops; the model reaches a steady state (i.e., the morphology of lens stays relatively stable) within 50 years. The model includes 22 layers with a depth of 50 m to improve on the vertical resolution in

the model simulation and is discretized into 41 columns in the horizontal direction with refinement at the boundaries. Constant head boundaries are defined along both sides of the domain to simulate the sea level, with density specified at 1025 kg/m$^3$, representative of typical seawater composition. Constant concentration boundaries are assigned to the same grid cells as the constant head boundaries with chloride concentrations of 19000 mg/L. Assuming that the loose sediments of the coral island are initially saturated by seawater, the initial chloride concentration of the entire model domain is specified at 19,000 mg/L. The ground surface is provided by the topographic map, and the lithology is based on boreholes. The model grid is shown in Figure 2.

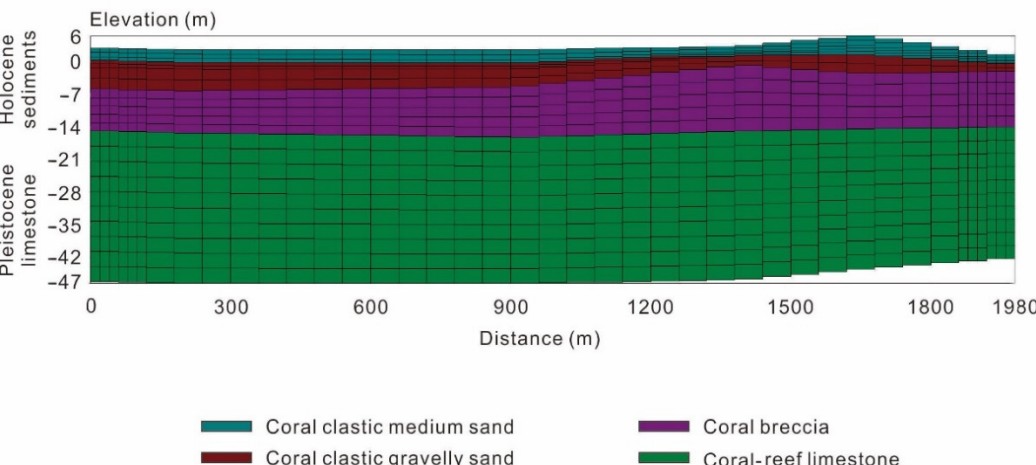

**Figure 2.** Model gird for the small coral island, divided into a mesh of 902 discrete rectangular elements, for performing simulations of groundwater flow and solute transport within the islet substrate.

Hydraulic conductivity of the principal aquifer is determined based on the lithology and previous studies [7,15]. Holocene sediments were divided into 12 model layers: 1–3 layers represent coral clastic medium sand, with the horizontal hydraulic conductivity ($K$) of 60 m/d; 4–7 layers represent coral clastic gravelly sand, with the horizontal hydraulic conductivity ($K$) of 110 m/d; 8–12 layers represent coral breccia, with the horizontal hydraulic conductivity ($K$) of 150 m/d. Pleistocene limestone, which has good permeability because of many pores and dissolution cavities of different sizes, includes 10 model layers (13–22), and the horizontal hydraulic conductivity ($K$) is 1000 m/d [26]. The effective porosity ($n_e$) is 0.25–0.45 and the specific yield ($S_y$) is 0.1–0.2.

Recharge is applied to the top layer of the model with a chloride concentration of 0 g/L to simulate the monthly average recharge to the aquifer. Recharge is the only input of freshwater to the hydrogeological system and, therefore, is the main mechanism by which the simulated freshwater lens develops in the model [27]. The monthly net recharge is calculated using the precipitation recharge coefficient ($\alpha$). Before the model reaches a steady state, the pumping wells are not activated. After the model reaches a steady state, the wells are activated for continuous pumping for 20 years to simulate the saltwater up-coning. The hydrogeological parameters assigned to the SEAWAT model are summarized in Table 1.

*3.2. Orthogonal Experimental Design*

The application of orthogonal experimental design is efficient, fast and economic, and has been widely used in many fields [28–31]. Orthogonal experimental design is an experimental design method to study multiple factors and multiple levels. According to orthogonality, some representative experimental scenarios are selected from the comprehensive experiments. When three or more components are involved in the experiment, and there may be interactions between them, the workload increases and gets difficult to

implement. Orthogonal experimental design is undoubtedly a better choice to solve this problem and can achieve equivalent results with a minimum number of tests.

**Table 1.** Parameter values used in the model for the coral island groundwater system.

| Settings | Parameter | Units | Value |
|---|---|---|---|
| Basic setup | Island width | m | 1980 |
| | Thickness | m | 50 |
| | Grid | \ | $41 \times 22$ |
| | Simulated time step | d | 36,525 |
| Flow model | Recharge | mm/y | Monthly average recharge |
| | Effective porosity | \ | 0.25–0.45 |
| | Holocene $K$ | m/d | 60–150 |
| | Pleistocene $K$ | m/d | 1000 |
| | Specific yield | \ | 0.1–0.2 |
| Transport model | Longitudinal dispersivity | m | 5 |
| Density-dependent model | Reference fluid density (Freshwater) | $kg/m^3$ | 1000 |
| | Seawater density | $kg/m^3$ | 1025 |

3.2.1. Orthogonal Table

The main tool of orthogonal experiment design is an orthogonal table. Before designing the orthogonal table, the experimenters need to determine the number and level of factors and make the factor-level table. Experiments generally involve multiple factors and levels, the most common being three factors and three levels. The factor-level table (three factors and three levels) format is shown in Table 2. The first column contains all the factors to be considered in the experimental design, and the values in the table are the specific values of each factor at each level. The design of the orthogonal table is based on the number and level of factors. Its principle is to select some representative experiments from all experiments according to the orthogonality. For example, three factors and three levels are required for $3^3$ = 27 trials following the traditional comprehensive experiments, and only nine trials are required after using the orthogonal table (L$_9$ ($3^3$)). In the three factors and three levels orthogonal table format, such as in Table 3, the first column represents the number of the experiments, and the specific values in the table represent the combination of different levels of factors. Orthogonality means that each level of one factor can meet once with each level of another factor. The orthogonal table has two characteristics:

- Different numbers in each column appear at the same time. For example, in the three factors and three levels orthogonal table, any column has 1, 2, and 3, and the number of occurrences in each column is equal.
- The arrangement of numbers in any two columns is comprehensive and balanced. For example, in the case of three levels, there are nine kinds of ordered number pairs in any two columns (in the same row), "1,1", "1,2", "1,3", "2,1", "2,2", "2,3", "3,1", "3,2", "3,3", and each pair appears the same times.

**Table 2.** Factor-level table (three factors and three levels).

| Factors | Levels | | |
|---|---|---|---|
| | 1 | 2 | 3 |
| Factor A | Value 1 | Value 2 | Value 3 |
| Factor B | Value 5 | Value 6 | Value 7 |
| Factor C | Value 8 | Value 9 | Value 10 |

**Table 3.** Orthogonal Table (three factors and three levels).

| Test Number | Levels | | |
|---|---|---|---|
| | **A** | **B** | **C** |
| 1 | 1 | 1 | 1 |
| 2 | 1 | 2 | 3 |
| 3 | 1 | 3 | 2 |
| 4 | 2 | 1 | 3 |
| 5 | 2 | 2 | 2 |
| 6 | 2 | 3 | 1 |
| 7 | 3 | 1 | 2 |
| 8 | 3 | 2 | 1 |
| 9 | 3 | 3 | 3 |

The above two points fully reflect the superiority of the orthogonal table, namely uniform dispersion and comparable.

### 3.2.2. Multi-index Range Analysis

Range analysis is an analysis method used for orthogonal experiment results, which is carried out after the data collection. Range ($R$) is the difference between the maximum and minimum values, which is used to represent the measures of variation in statistical data. Range analysis determines the primary and secondary order of different factors affecting the experimental results by comparing $R$. The larger the $R$, the larger the swing range of the factor in different levels of data, indicating that the factor has a more significant impact on the results. Range analysis results can be recorded in the range analysis table (Table 4). If there are multiple indicators, range analysis is required for each indicator. The operation process is as follows. Firstly, the experiment results of each factor at the same level of a single indicator are summed, respectively, and the sum obtained is denoted as $K$. Then, the mean values of $K1$, $K2$, and $K3$ are calculated, respectively, which are denoted as $\bar{K}1$, $\bar{K}2$, $\bar{K}3$, and the optimal level of each factor is determined by $\bar{K}$. $R$ is then calculated and the importance of factors are sorted accordingly. Finally, the above operation is repeated for each indicator. For example (Table 4), for factor A, when the level is 1, $\bar{K}$ is the largest, which means level 1 is the optimal level of factor A for indicator 1. Since $RB > RA > RC$, the influence of factors on indicator1 is ranked as BAC.

**Table 4.** Range analysis table.

| | Items | Factors | | |
|---|---|---|---|---|
| | | **A** | **B** | **C** |
| | $K1$ | Value A1 | Value B2 | Value C3 |
| | $K2$ | Value A2 | Value B2 | Value C2 |
| | $K3$ | Value A3 | Value B3 | Value C3 |
| | $\bar{K}1$ | Value A1/3 | Value B1/3 | Value C1/3 |
| Indicator | $\bar{K}2$ | Value A2/3 | Value B2/3 | Value C2/3 |
| | $\bar{K}3$ | Value A3/3 | Value B3/3 | Value C3/3 |
| | optimization level of factors | A1 | B3 | C3 |
| | $R$ | max$\bar{K}$-min$\bar{K}$ | max$\bar{K}$-min$\bar{K}$ | max$\bar{K}$-min$\bar{K}$ |
| | Importance ranking | | BAC | |

### 3.2.3. Comprehensive Balance Analysis

The comprehensive balance analysis method is to investigate the influence of each factor on each index, after which analysis and comparison can be performed to determine the best level, to obtain the best test scheme. The operation process is as follows. Firstly, the multi-index range analysis is carried out to obtain a better combination of several factors. In this process, the primary and secondary factors of each indicator are identified, and the level of each factor is determined. Then, the overall optimal factor level combination is

determined based on weighing the impact on other indicators when the optimal level of a factor is selected.

## 4. Results and Discussion

### 4.1. Pumping Wells Layout Scheme Design of the Freshwater Lens

"Sustainable exploitation" is an extension of "allowable exploitation", based on the concept of sustainable development (UNCED, 1992) [32]. In 1999, Alley et al. [33] defined sustainable groundwater development as "long-term and permanent development and utilization of groundwater without serious social, economic and environmental consequences". The concept of sustainable exploitation highlights environmental factors and emphasizes the renewable and sustainable utilization of groundwater resources. This study proposes that the essence of sustainable exploitation for the groundwater system of coral islands is that after a change in source and sink terms due to human intervention, the freshwater lens system changes from the original dynamic equilibrium to a new dynamic equilibrium, and the latter is acceptable. Because the variation in the groundwater table is at centimeter-level and the variation in the concentration isoline is at meter level in many scenarios of this simulation, this study regards the saltwater up-coning height as the main limiting factor of sustainable exploitation (Figure 3).

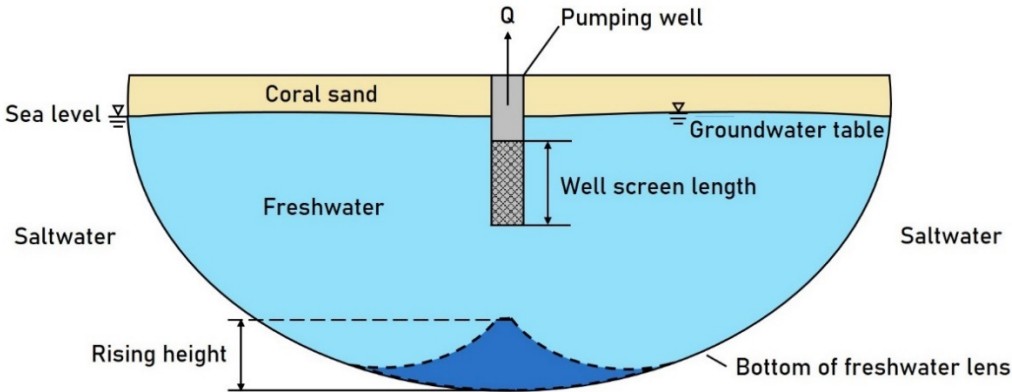

**Figure 3.** Schematic diagram of the saltwater up-coning process.

A pumping well extracting from the freshwater lens can cause the saltwater to move upwards towards the well, a phenomenon known as saltwater up-coning [6,34]. A schematic diagram of the saltwater up-coning process is shown in Figure 4. In this study, when the groundwater numerical model reaches the steady-state, the corresponding pumping rate when the rising height just reaches the bottom of the well screen is defined as the critical pumping rate ($Q_C$). The sum of $Q_C$ of all pumping wells can then be expressed by the total critical pumping rate ($Q_T$), and the sustainable yield cannot exceed $Q_T$. $T_m$ is the maximum thickness of the freshwater lens under pumping conditions, and $T_C$ is the central thickness (Figure 4). The coordinate corresponding to the $T_m$ changes with pumping conditions, but the position corresponding to the $T_C$ will not change. Due to pumping, the $T_C$ doesn't coincide with the $T_m$. In this study, the length of the well screens, the number of wells, and the distance between wells are selected as the factors of the design scheme of pumping wells, and each factor has three levels (Table 5). One of the typical well layout schemes is shown in Figure 4.

In this study, we use a chloride concentration of 250 mg/L as the upper threshold for fresh groundwater, which is the World Health Organization (WHO) [35] recommendation for drinking water in 2011. However, other researchers have also used 500 mg/L and 600 mg/L as the upper threshold of chloride concentration [14,36,37]. When the model runs stably without pumping, the freshwater flow is stressed by saltwater when the boundary is moved on both sides, whereas saltwater moves upward driven by buoyant effects in the middle of the model [38]. Three variables are selected as evaluating indicators in the

numerical test: total critical pumping rate ($Q_T$), the central thickness of the freshwater lens ($T_C$), and maximum thickness of the freshwater lens ($T_m$) (Figure 4). The orthogonal experimental design scheme and its corresponding numerical test results are shown in Figure 5 and Table 6.

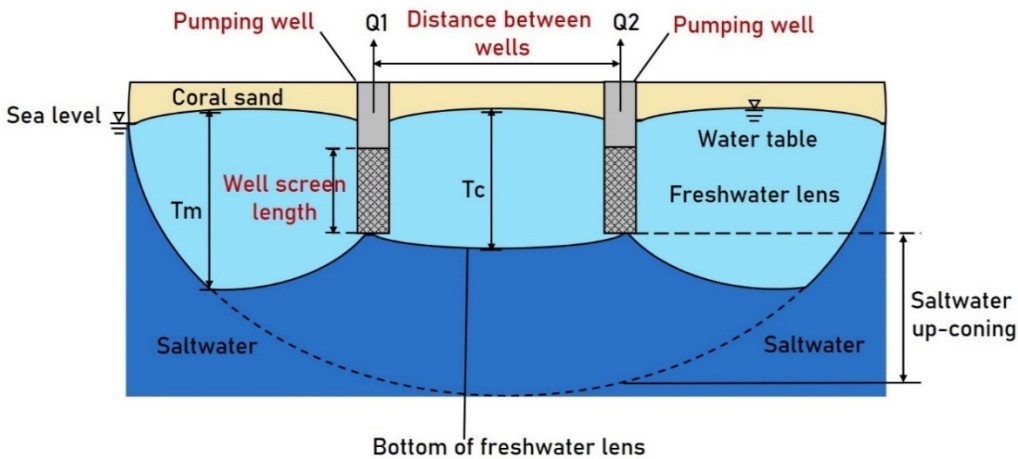

**Figure 4.** Schematic diagram of the linear well layout scheme.

**Table 5.** Factor-level table of well layout plan (3 factors and 3 levels).

| Factors | Levels | | |
|---|---|---|---|
| | 1 | 2 | 3 |
| A: Screen length | 2 m | 3 m | 4 m |
| B: Number of wells | 2 | 4 | 6 |
| C: Distance between wells | 100 m | 150 m | 200 m |

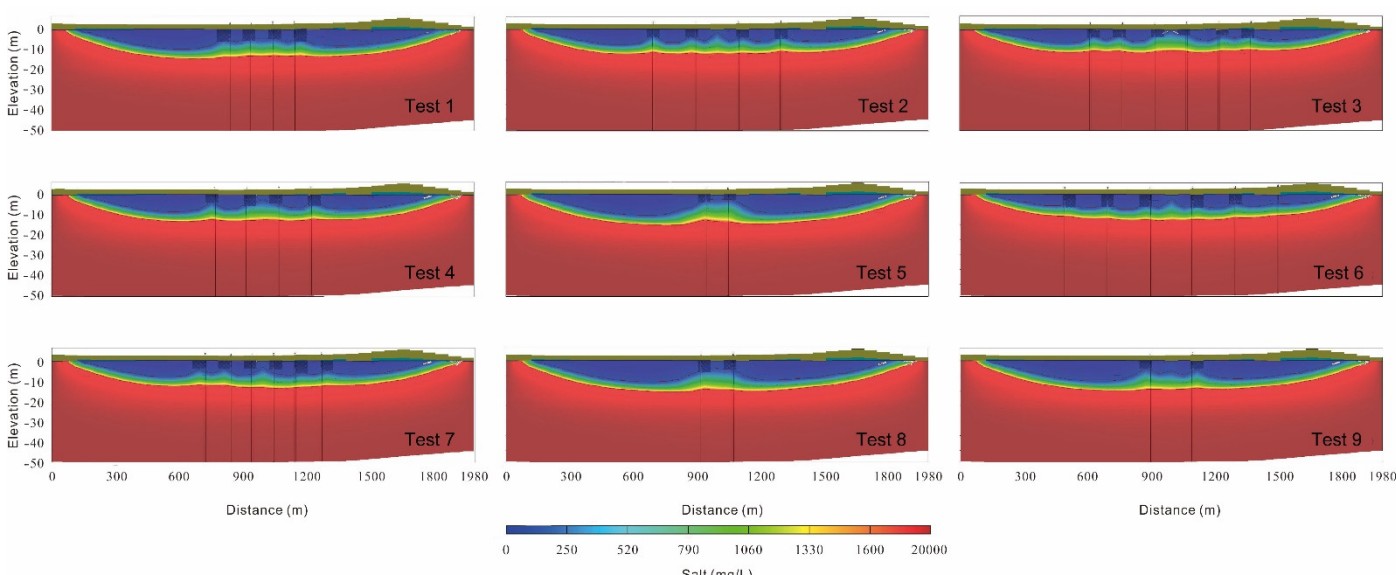

**Figure 5.** Distribution of salinity in cross-section of different experimental scenarios.

**Table 6.** Orthogonal experimental design and results.

| Test Number | Levels | | | Test Results | | |
|---|---|---|---|---|---|---|
| | **A** | **B** | **C** | $Q_T$ (m³/d) | $T_C$ (m) | $T_m$ (m) |
| 1 | 3 | 2 | 1 | 0.407 | 4.30 | 9.20 |
| 2 | 1 | 2 | 3 | 0.730 | 1.10 | 7.40 |
| 3 | 1 | 3 | 2 | 0.907 | 0.50 | 6.10 |
| 4 | 2 | 2 | 2 | 0.555 | 2.30 | 8.20 |
| 5 | 1 | 1 | 1 | 0.330 | 2.20 | 9.80 |
| 6 | 3 | 3 | 3 | 0.735 | 1.70 | 7.00 |
| 7 | 2 | 3 | 1 | 0.562 | 2.80 | 7.50 |
| 8 | 3 | 1 | 2 | 0.215 | 3.30 | 10.00 |
| 9 | 2 | 1 | 3 | 0.330 | 4.60 | 9.80 |

The numerical experiment results show that different pumping well layout schemes have a great impact on $Q_T$. The $Q_T$ of test 3 is 4.2 times that of test 8 (Table 6). Of course, $Q_T$ is not the only factor to judge whether the scheme is the best or not. It is also necessary to comprehensively analyze the thickness of the freshwater lens under the pumping scheme. If the thickness of the freshwater lens becomes too shallow, the current pumping scheme is not reasonable, although a large pumping rate may be obtained. In principle, the sustainable yield shall not exceed $Q_T$. The determination of $Q_T$ can provide a reference for the decision-making of sustainable yield.

According to the calculation results and difference analysis of orthogonal tests (Table 7), $RB > RA > RC$ for $Q_T$ and $T_m$, $RA > RB > RC$ for $T_C$. The results indicate that the number of wells, and the length of well screens, are the most critical factors in the design of the well layout scheme, followed by the distance between wells. Although the more wells, the greater the $Q_T$, if the number of wells is increased without considering the distance between wells, the $T_C$ will be greatly reduced and the risk of damage to the freshwater lens will increase. Therefore, the number of wells needs to be limited. The longer the well screen and the deeper the buried depth at the bottom of the well screen, the smaller the $Q_T$, so the design of the length of the well screen needs to be cautious. If long-term pumping is implemented under the conditions of short well screens with $Q_T$, it is likely to reduce the $T_C$ to less than 1/3 of the original thickness, which increases the risk of damage to the freshwater lens. Therefore, it is necessary to control the pumping intensity of shallow wells.

In order to reduce $T_C$ and $T_m$ as little as possible and obtain a larger $Q_T$, the comprehensive balance analysis method is used to select the optimization scheme. Compared with $Q_T$ and $T_m$, factor A has a more significant influence on $T_C$. It can be seen from Table 4. that A2 is the best choice. The influence of A on $Q_T$ and $T_m$ occupies second place among the three factors, with A1 and A3 being the best choices. However, when A1 is selected, $T_C$ and $T_m$ are very thin, so A1 is not considered; when A2 is selected, $T_m$ is 2.6% less than that of A3; when A3 is selected, not only is $T_C$ reduced by 4%, but $Q_T$ is also reduced by 6% compared with A2, so A2 is considered comprehensively. The influence of factor B on $Q_T$ ranks first among A, B, and C. However, when B3 is taken, $T_C$ is less than 1/3 (2.5 m) of the original thickness. When B2 is taken in turn, there is the same problem, so B1 is taken finally, which is also the best level for $T_C$. According to Table 4, the influence of factor C on $Q_T$, $T_C$, and $T_m$ takes third place, and C1 and C3 can be taken into consideration. When C1 is selected, $T_C$ and $T_m$ increase by 25.5% and 9.4%, respectively; When C3 is selected, $Q_T$ will increase by 39.5% compared with C1, but $T_C$ and $T_m$ are only reduced by 20.3% and 8.6%, respectively, so C3 is a better choice. As a result, the best scheme is A2B1C3, that is, the linear well layout scheme with a 3 m well screen length, two wells, and 200 m of the distance between wells (Test 9). This distance accounts for about 10% of the island width. Under this scheme, $Q_T$ is 0.33 m³/d. $T_C$ is 4.6 m, which accounts for 60% of the original thickness, and $T_m$ is 9.8 m, which accounts for 86% of the original thickness.

**Table 7.** Multi-index range analysis of orthogonal experimental results.

| | Items | Factors | | |
|---|---|---|---|---|
| | | **A** | **B** | **C** |
| $Q_T$ | K1 | 1.97 | 0.88 | 1.30 |
| | K2 | 1.45 | 1.69 | 1.68 |
| | K3 | 1.36 | 2.20 | 1.80 |
| | $\bar{K}1$ | 0.66 | 0.29 | 0.43 |
| | $\bar{K}2$ | 0.48 | 0.56 | 0.56 |
| | $\bar{K}3$ | 0.45 | 0.73 | 0.60 |
| | Optimization level of various factors | A1 | B3 | C3 |
| | R | 0.20 | 0.44 | 0.17 |
| | Primary and secondary order | | BAC | |
| $T_c$ | K1 | 3.80 | 10.10 | 9.30 |
| | K2 | 9.70 | 7.70 | 6.10 |
| | K3 | 9.30 | 5.00 | 7.40 |
| | $\bar{K}1$ | 1.27 | 3.37 | 3.10 |
| | $\bar{K}2$ | 3.23 | 2.57 | 2.03 |
| | $\bar{K}3$ | 3.10 | 1.67 | 2.47 |
| | Optimization level of various factors | A2 | B1 | C1 |
| | R | 1.97 | 1.70 | 1.07 |
| | Primary and secondary order | | ABC | |
| $T_m$ | K1 | 23.30 | 29.60 | 26.50 |
| | K2 | 25.50 | 24.80 | 24.30 |
| | K3 | 26.20 | 20.60 | 24.20 |
| | $\bar{K}1$ | 7.77 | 9.87 | 8.83 |
| | $\bar{K}2$ | 8.50 | 8.27 | 8.10 |
| | $\bar{K}3$ | 8.73 | 6.87 | 8.07 |
| | Optimization level of various factors | A3 | B1 | C1 |
| | R | 0.97 | 3.00 | 0.77 |
| | Importance ranking | | BAC | |

Note: K1 corresponding to factor A is the sum of $Q_T$ corresponding to level 1; $\bar{K}1$ is the mean value of K1 at level 1. R is range (max$\bar{K}$-min$\bar{K}$).

### 4.2. Uncertainty Analysis of $Q_T$ and Main Influence Factors

Due to the uncertainty of hydrogeological parameters, $Q_T$ in the above-mentioned pumping wells layout scheme is also uncertain. Hydraulic conductivity ($K$), vertical dispersion coefficient ($DI$), precipitation infiltration coefficient ($\alpha$), effective porosity ($n_e$), and specific yield ($S_y$) are the influencing factors:

$$Q_T = F(K, DI, \alpha, n_e, S_y) \tag{1}$$

Sensitivity analysis is used to analyze the uncertainty of $Q_T$ and identify the main risk factors. Sensitivity can measure the impact of the change in influencing factors on the indicators. The global sensitivity analysis can fully account for the effect of parameter interaction on the simulation results. In this study, the method proposed by Morris [39] in 1991 was adopted. By adjusting only one parameter at a time, the influence of each parameter on the overall results is calculated in turn, and the importance of each parameter is effectively prioritized. The influence of parameter changes on the overall results is measured by sensitivity ($S_i$), expressed as:

$$S_i = \frac{(y_i - y_0)/y_0}{(X_i - X_0)/X_0} \tag{2}$$

where $S_i$ is the sensitivity of function y(x) to variable $X_i$; $y_0$ is the value of $Q_T$ before the parameter changes; $X_0$ is the initial parameter. Among them, the larger the value of $|S_i|$ is, the more sensitive the variable $X_i$ is, and the greater the influence on the function is.

When $K$, $DI$, $\alpha$, $n_e$, and $S_y$ change from 0 to 30% respectively, the $|S_i|$ of $\alpha$ is the largest, ranging from 0 to 2.020, therefore, changes in $\alpha$ will have a greater impact on $Q_T$ compared with other four parameters with the same amplitude. $|S_i|$ of $n_e$ is in the second place, ranging from 0 to 0.566, and the influence of $n_e$ is much smaller than that of $\alpha$. The other three parameters have less influence, so the focus should be on the changes in $\alpha$ and $n_e$. When $K$, $DI$, $\alpha$, $n_e$, and $S_y$ change from $-30\%$ to 0, the $|S_i|$ of $\alpha$ and $K$ are significantly greater than the other three parameters, ranging from 0 to 1.202 and 0 to 0.949, respectively. Therefore, the focus should be on the changes of $\alpha$ and $K$ on the determination of $Q_T$. In conclusion, from $-30\%$ to 30%, $\alpha$, $K$, and $n_e$ are the main influencing factors in the determination of $Q_T$.

The results of the global sensitivity analysis of $\alpha$–$K$, $\alpha$–$n_e$, $K$–$n_e$ and $\alpha$–$K$–$n_e$ show that the comprehensive influence of $\alpha$–$n_e$ is the largest, and the corresponding $|S_i|$ ranges from 0 to 2.182. When the parameter amplitude is $-30\%$ to 0, the comprehensive influence of $\alpha$ and $n_e$ on $Q_T$ is less than the sum of their respective influence, but when the parameter amplitude is 0 to 30%, the comprehensive influence of $\alpha$–$n_e$ on $Q_T$ is greater than the sum of their respective influence. The analysis results of local sensitivities of all parameters and joint sensitivities of main influencing parameters are shown in Table 8 and Figure 6.

**Table 8.** Calculation results of parameter sensitivity.

| Percentage of Parameters Changes Parameters | −30% | −15% | 0% | 15% | 30% |
|---|---|---|---|---|---|
| | | | $|S_i|$ | | |
| $DI$ | 0.202 | 0.364 | 0.000 | 0.323 | 0.192 |
| $S_y$ | 0.172 | 0.343 | 0.000 | 0.343 | 0.182 |
| $n_e$ | 0.101 | 0.364 | 0.000 | 0.566 | 0.293 |
| $\alpha$ | 1.202 | 0.869 | 0.000 | 2.020 | 1.667 |
| $K$ | 0.717 | 0.949 | 0.000 | 0.202 | 0.313 |
| $\alpha$–$K$ | 0.788 | 0.485 | 0.000 | 1.616 | 1.222 |
| $\alpha$–$n_e$ | 1.242 | 0.970 | 0.000 | 2.182 | 1.828 |
| $K$–$n_e$ | 0.523 | 0.395 | 0.000 | 0.028 | 0.046 |
| $\alpha$–$K$–$n_e$ | 0.899 | 0.566 | 0.000 | 1.717 | 1.354 |

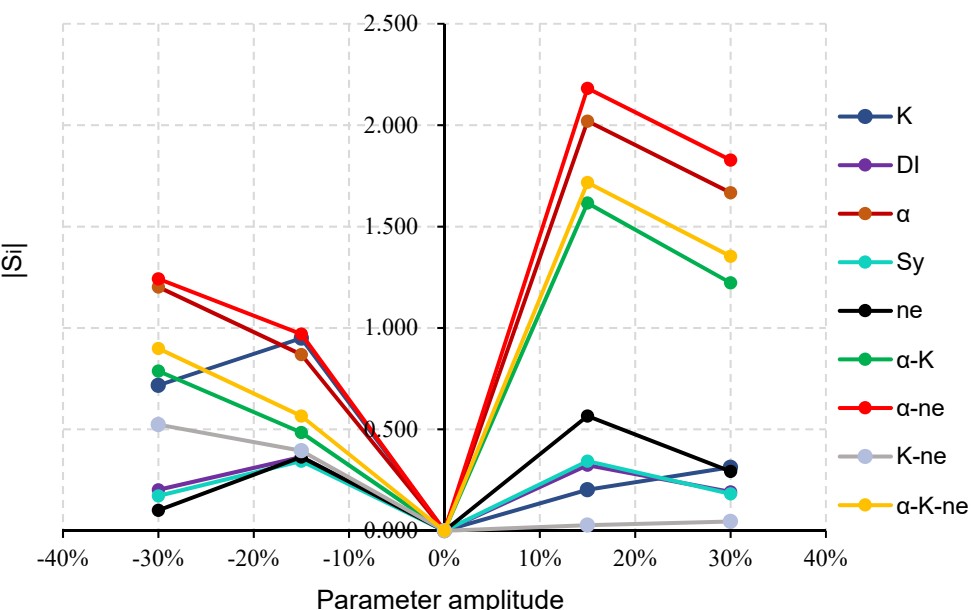

**Figure 6.** Parameter sensitivity analysis diagram.

The increase in $\alpha$ and $n$ has positive impacts on $Q_T$, but the increase in $K$ has negative impacts (Figure 7). The reason for this is that $\alpha$ affects the amount of the net recharge. A

larger $\alpha$ means that the freshwater lens can obtain a larger hydraulic gradient to maintain or even increase the thickness of freshwater, and has a larger maximum permissible rising height of saltwater, resulting in a larger $Q_T$. With the increase in $K$, the thickness of the freshwater lens will become thinner, and the maximum allowable rising height of saltwater will decrease, which requires pumping rates to be relatively reduced to protect the wells from salinization.

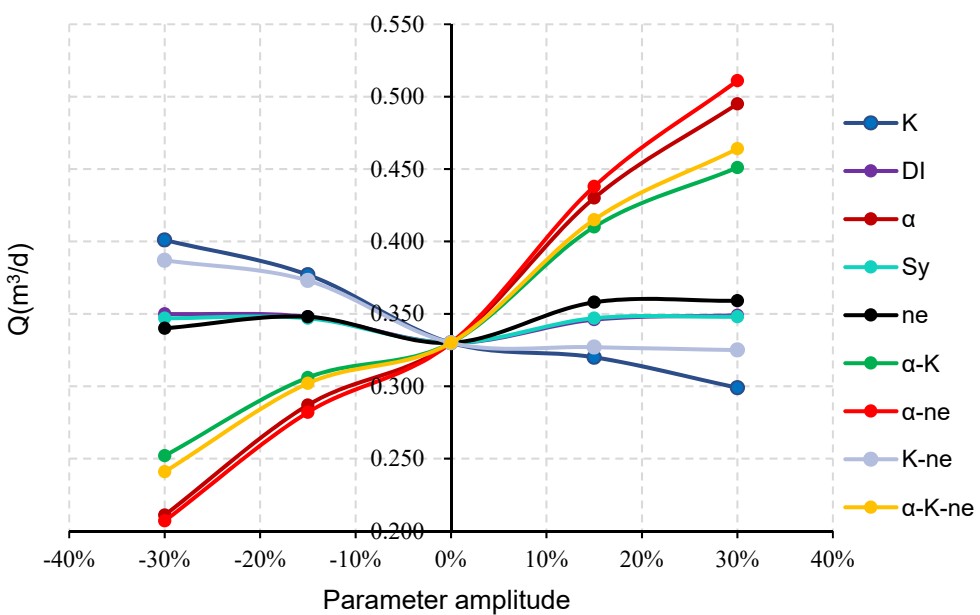

**Figure 7.** The variations of $Q_T$ for different parameter combinations.

Under the most unfavorable parameter combination ($\alpha$ and n decrease by 30%, $K$ increases by 30%), $Q_T$ is 0.174 m$^3$/d, which is only 52.7% of the original (0.33 m$^3$/d), so the original calculated $Q_T$ is not absolutelyreliable; under the most favorable parameter combination ($\alpha$ and $n$ increases by 30%, $K$ decreases by 30%), $Q_T$ is 0.573 m$^3$/d, which is 73.63% higher than the original $Q_T$. Therefore, the range of $Q_T$ is from 0.174 to 0.573 m$^3$/d. The identification of the main influencing factors can help identify the key risk sources that affect the determination of $Q_T$, from the standpoint of hydrogeological parameters. The results show that the key risk sources of $Q_T$ are the uncertainties of $\alpha$, $n$, and $K$. The influence of parameters on the determination of $Q_T$ is related to the variation range of the parameters, within the range of a 30% reduction in parameters, $\alpha$ is the most important risk factor of pumping, and $K$ is the second; within the range of a 30% increase in parameters, $\alpha$ is also the most important risk factor, followed by $n$ and $K$. The uncertainty of the hydrogeological parameters has a great impact on $Q_T$, but their spatial variabilities haven't been taken into consideration, which may also influence the results, and is meant for relevant analysis.

The 2D model assumes that recharge is stored in the profile and there is no flow movement perpendicular to the profile. This generalization may lead to a slight overestimation of reserves. However, the overestimation is generally within a reasonable range. Numerical models can be used as a powerful tool to predict and evaluate the water resources of coral islands, but it is difficult to avoid errors in comparison to reality, especially in the conceptualization of hydrogeological conditions of coastal aquifers. The determination of sustainable yield should be combined with the specific well layout schemes, and the uncertainty of hydrogeological parameters should also be taken into account, because of the complex hydrogeological conditions of coral islands.

## 5. Summary and Concluding Remarks

Groundwater resources of coral islands are critical for water supply and have great economic and social value. This study provides an optimal well layout scheme by using

the orthogonal experimental design and numerical simulation and analyzes the sensitivity of the pumping rate to the uncertainty of hydrogeological parameters. On this basis, the main influencing factors affecting the determination of pumping rates are selected, and the possible risks associated with the uncertainty of hydrogeological parameters are suggested. The results show that:

(1) The determination of $Q_T$ should be based on the specific well layout, with consideration of the length of well screens, the number of wells, and the distance between wells, because the calculation results of $Q_T$ corresponding to different well layouts can differ by three times.

(2) The orthogonal experimental design is adopted to analyze the important elements of well layout, and thereby the results based on multi-index extremum range analysis and comprehensive balance analysis show that the number of wells and the length of well screens are the most critical factors in the design of well layout scheme, followed by the distance between wells.

(3) In this study, the optimal well layout is selected from nine schemes. The length of the well screen of the scheme is 3 m, the number of wells is two and the distance between wells is 200 m, which accounts for about 10% of the island width. For the optimal scheme, $Q_T$ is 0.33 m$^3$/d, $T_C$ is 4.6 m, which accounts for 60% of the original thickness, and $T_m$ is 9.8 m, which accounts for 86% of the original thickness.

(4) The sensitivities of the hydrogeological parameters with respect to $Q_T$ are significant.

Within the range of a 30% reduction in parameters, $\alpha$ and $K$ are the key risk factors of pumping; within the range of a 30% increase in parameters, $\alpha$, $n$ and $K$ are the key risk factors; $\alpha$–$n$ combined changes had the greatest impact. Under conditions of the most unfavorable parameter combination, $Q_T$ is only 52.7% of the original value.

The management of freshwater lenses and the assessment of sustainable yields, together with the uncertainty of hydrogeological factors, will continue to be the primary activity of coral islands. This paper only focuses on the variation and combination of parameters, without considering the spatial variability of hydrogeological parameters in the basic model. Spatial variability of the hydrogeological parameters may also have an important influence on the optimization of well layout and the sensitivity of pumping rate and is recommended for future work.

**Author Contributions:** Conceptualization, R.W. and L.S.; methodology, R.W., software, R.W., writing—original draft preparation, R.W.; writing—review and editing, R.W., L.S., Y.L. and P.A.O.; supervision, L.S.; project administration, L.S.; funding acquisition, L.S. All authors have read and agreed to the published version of the manuscript.

**Funding:** This research was funded by the Central University Project "Risk Analysis of the Developmentand Utilization of Island Freshwater Lenses" (No. B200203046), the Postgraduate Scientific Researchand innovation Plan of Jiangsu Provence (No. kycx20_0461), and the Major Innovation and Technology Projects of Shandong Province (No. 2019JZZY020105).

**Data Availability Statement:** Data is contained within the article.

**Acknowledgments:** All authors are very grateful to the editor and the anonymous reviewers for their valuable comments, which have greatly improved the paper.

**Conflicts of Interest:** The authors declare no conflict of interest.

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
