# Peer review of "Pumping Well Layout Scheme Design and Sensitivity Analysis of Total Critical Pumping Rates in Coral Island Based on Numerical Model"

_water, doi:10.3390/w13223215_

Round 1
Reviewer 1 Report
The manuscript presents an interesting hydrogeological modeling exercice on the case of freshwater lens in small island context. Such an exercice has practical implications for the sustainable exploitation of this type of water resource. It is rather well written and illustrated.
I have provided comments and remarks directly in the annotated manuscript.
In my opinion, the main problem is the presentation of the orthogonal experimental design approach. It seems quite a novel approach in its application to the field of groundwater flow modeling, therefore the methodology shall be explained in more detail. A similar remark holds for the Multi-index Extremum Difference Analysis. More details on the methodology will help the reader understands the outcome of such an analysis.

Author Response
Thank for your comments concerning our paper. We have made careful corrections that we hope to meet your approval. Please see the attachment.

Reviewer 2 Report
The paper presents a numerical study of the effect of multiple pumping well on small island's coastal aquifer.
First, the paper is misleading. The tile of the paper must say that the authors do only numerical modeling.
Then, the paper is a very simplistic numerical modeling exercise.
The paper does not consider spatial variability of the hydraulic conductivity within the geological units which is very limiting in terms of interpretation of the effects of pumping.
The choice of parameters is obsolete. Why not using a full Markov process as running small 2D or 3D numerical models is not time consuming?
The part on the description of parameter R is not clear.
Equation 1 is not necessary ( and is wrong as the sum contains the indices, i.e., not necessary to addd (i=1,2, ..., N))
Figure quality is poor.
The conclusions are obvious.
English must be corrected. Some sentences have not the same meaning in english:
- Previous research on the freshwater lens has been conducted by scholars at home and abroad.
- Extremum difference (R) refers to the difference between the maximum value and the minimum value of the average value of the test index corresponding to each level in each column.
A large literature exist in real experiments on coastal aquifers. For example:
- Challenges in groundwater resource management in coastal aquifers of East Africa: Investigations and lessons learnt in the Comoros Islands, Kenya and Tanzania
- Groundwater prospection in Grande Comore Island - Joint contribution of geophysical methods, hydrogeological time-series analysis and groundwater modelling
Author Response
Thank you for your comments on our paper. We have made careful corrections and hope to get your approval. Please see attachment.

Reviewer 3 Report
This manuscript reports a interesting work.
I attach a file with some improving suggestions

Author Response
Thanks for your comments concerning our paper. We have made careful corrections that we hope to meet your approval. Please see the attachment.
